# A novel strategy for resource utilization of oily drilling waste fluids in northern Shaanxi: Stepwise flotation of bentonite and barite using SDS and interfacial reaction mechanisms

Liu Lihu[1], Shi Jiahe [ID][2]*

1 Wuqi Oil Production Plant, Yanchang Oilfield Co., Ltd, Yan'an, China, 2 Bristol Business School, University of Bristol, Bristol, United Kingdom

* shijiahe1917@163.com

## Abstract

Northern Shaanxi's oil-gas drilling produces large amounts of waste drilling fluids with high-value solids (barite, bentonite). Traditional disposal causes resource waste and pollution. This study proposes a stepwise flotation process for typical local oil-based waste: surface cleaning to break oil film wrapping and combined reagents to regulate mineral surface hydrophobicity differences, enabling efficient separation and recovery of barite and bentonite. The flotation mechanism is also preliminarily explored. The experimental results show that: ultrasonic cleaning with 0.5% sodium dodecyl sulfate (SDS) solution makes the oil desorption rate of solid phase >95%, restoring the natural surface properties of minerals. Through the stepwise flotation design, in the first stage at pH = 4.0, 0.8 kg/t sodium dodecyl sulfate and 0.6 kg/t sodium hexametaphosphate are added, and the recovery rate of bentonite reaches 86.3%; in the second stage at pH = 8.0, 1.2 kg/t sodium dodecyl sulfate and 0.7 kg/t gellan gum are added, and the grade of barite concentrate is 92.1% ($BaSO_4$ content), with a recovery rate of 88.7%. Through the flotation closed-circuit experiment, using the process of "one roughing, two cleaning and three scavenging", the first stage can obtain bentonite concentrate with a recovery rate of 91.4% and a grade of 91.5%; the second stage can obtain barite concentrate with a recovery rate of 90.2% and a grade of 92.1%. SEM shows that bentonite is dissociated lamella, and barite presents clean prismatic crystals without oil film impurities, verifying the high efficiency of separation. Mechanism studies show that SDS has dual functions of oil breaking and collecting. Sodium hexametaphosphate inhibits the flotation of barite by chelating $Ba^{2+}$ in barite. Gellan gum realizes separation by shielding the active sites of bentonite through hydrogen bonds. This study provides an efficient and low-consumption solution for the resource utilization of drilling waste liquid in northern Shaanxi.

**Data availability statement:** All relevant data are within the manuscript.

**Funding:** The author(s) received no specific funding for this work.

**Competing interests:** The authors have declared that no competing interests exist.

# 1 Introduction

Northern Shaanxi, as an important energy and chemical industry base in China, sees intensive oil-gas drilling activities generating over 500,000 $m^3$ of waste drilling fluids annually [1–4]. These fluids are rich in high-value solid components, including barite ($BaSO_4$) as a drilling fluid weighting agent (density 4.0–4.5 $g/cm^3$), accounting for 15–30%, and bentonite as a viscosifier (montmorillonite content >60%), accounting for ~10–20%. Their effective recovery can significantly reduce the pressure on mineral exploitation— the grade of barite ores around Northern Shaanxi has gradually decreased to 80%, while the import dependency of bentonite reaches up to 40% [5–8]. However, local drilling fluids are mainly oil-based systems (oil content 8–15%), and mineral particles are tightly wrapped by oil films, making it difficult for traditional solid-liquid separation technologies to achieve efficient resource recovery [9].

Currently, the drilling waste fluids in Northern Shaanxi are mainly treated by solidification landfilling or formation reinjection. The former consumes large amounts of cement (200–300 $kg/m^3$) and causes conflicts with agricultural and forestry land due to land occupation; the latter poses potential risks of groundwater pollution (migration risks of heavy metals Pb/Cr) [10–12]. More severely, the problem of resource waste is prominent: each ton of waste fluid incurs a loss of ~280 CNY for barite and ~150 CNY for bentonite. Although mineral recovery technologies such as magnetic separation and gravity separation have been explored, the separation efficiency for oil-coated solids and fine-grained minerals (−20 μm accounting for >45%) is both below 50%, failing to meet industrial requirements [13].

There are three major shortcomings in the current mineral recovery research: (1) insufficient oil film removal: thermal desorption method (>400°C) has high energy consumption (cost ≥600 CNY/ton), while solvent cleaning (such as toluene) causes secondary pollution; (2) poor separation selectivity: the density of barite and bentonite overlaps (2.5–4.5 $g/cm^3$), making it difficult for conventional technologies to separate them; (3) low reagent compatibility: anionic/cationic collectors lose selectivity under oil phase interference [14–16]. To address the above bottlenecks, this study proposes a synergistic process of "surface cleaning-stepwise flotation", with innovations embodied in: (1) green pretreatment: 0.5% SDS solution is used for ultrasonic cleaning to strip the oil film and restore the natural surface properties of minerals; (2) combined reagent regulation: in the first stage at pH = 4.0 (adjusted by $H_2SO_4$), 0.8 kg/t sodium dodecyl sulfate (SDS) is used as the bentonite collector, and 0.6 kg/t sodium hexametaphosphate (SHMP) inhibits barite flotation; in the second stage at pH = 8.0 (adjusted by $Na_2CO_3$), 1.2 kg/t SDS captures barite, and 0.7 kg/t gellan gum selectively inhibits residual bentonite. The process is verified by a flotation closed-circuit process of "one roughing, two cleaning and three scavenging", and the flotation mechanism is explored. The innovative stepwise flotation of this work, combined with the introduction of the novel inhibitor gellan gum technology into drilling waste fluid flotation, solves the problem of selective separation of oil-based minerals, providing an efficient and low-consumption solution for the resource utilization of drilling waste fluids in northern Shaanxi.

## 2 Materials and methods

### 2.1 Materials

The waste drilling fluid was collected from an oil production plant in northern Shaanxi (with the density of the original waste drilling fluid being 1.4–1.6 g/cm³). The sample was dried, ground, and sieved through a 200-mesh sieve, and the sieve residue was selected for XRF(X-ray Fluorescence Spectrometry, XRF) quantitative analysis, with the results shown in Table 1. Barite ($BaSO_4$), as the main target mineral for recovery in this study, accounts for 20.1% of the solid phase.

The contents of Ba (11.8%) and S (4.5%) are consistent with the stoichiometric ratio of $BaSO_4$ [theoretical value of Ba:S = 58.84:13.73. Calculation method: Based on the molar mass of $BaSO_4$ (233.39 g/mol), the atomic weight of Ba is 137.33 g/mol (accounting for $137.33/233.39 \approx 58.84\%$), and the atomic weight of S is 32.07 g/mol (accounting for $32.07/233.39 \approx 13.73\%$)], verifying the reliability of barite as a core resource for recovery. The overall composition of the solid phase is dominated by quartz (Si 15%, equivalent to $SiO_2$ of approximately 32%), carbonate (Ca 10%, equivalent to calcite of approximately 25%), and bentonite (Al 8%, equivalent to montmorillonite of approximately 21%), with the total proportion of these three exceeding 78% of the total solid phase.

### 2.2 Equipment

Single-Trough Flotation Machine XFD-1L, Shandong Xinhai Mining Technology & Equipment Inc., China; Electronic Balance, Shanghai Wenying Electromechanical Equipment Co., Ltd., China; Nanoparticle & Zeta Potential Analyzer Litesizer™ 500, Anton Paar Instruments (Beijing) Co., Ltd., China; Wavelength Dispersive X-ray Fluorescence Spectrometer (WDXRF), PANalytical B.V. (Netherlands); Optical Contact Angle Goniometer OSA200, Ningbo New Border Scientific Instruments Co., Ltd., China; Field Emission Scanning Electron Microscope (FE-SEM) S-4800, Hitachi High-Technologies Corporation (China), Ltd; Laboratory Small-Scale Pulverizer BF-10, Hebei Bochen Technology Co., Ltd., China; Portable pH Meter PHB-4, Beijing Airan Technology Co., Ltd., China; X-Ray Diffractometer (XRD) D/max 2500, Rigaku Corporation, Japan.

### 2.3 Methods

The waste drilling fluid samples used in this study were obtained from the waste drilling fluid generated during a drilling operation at Zhidan Oil Production Plant of Yanchang Oilfield. The acquisition of the samples used in the experiment has obtained permission from Yanchang Oilfield. The chemical reagents involved in the experiment (such as HCl and inhibitors) were all purchased through compliant channels and comply with laboratory safety management regulations.

**2.3.1 Flotation experiment of waste drilling fluid.** 200 g of high-density drilling fluid samples were placed in a flotation cell, and 800 mL of deionized water was added. The pH value of the slurry was adjusted using a 10% (by volume) HCl solution. After conditioning the slurry for 10 minutes, add the depressant, frother, and collector in sequence, followed by aeration flotation at 2000 rpm for 6 minutes. The flotation experiment procedure is illustrated in Fig 1. The concentrate products obtained from the flotation cell were dried and weighed, and the recovery rate was then calculated.

**2.3.2 XRF testing.** XRF testingwas performed in accordance with *General Rules for X-ray Fluorescence Spectrometric Analysis (GB/T 16597−2019) and instrument operating specifications: The solid residue of waste drilling fluid passing through a 200-mesh sieve was taken and dried at 105°C for 2 hours; 5.0 g of the dried sample was weighed and ground in an agate mortar to a particle size of ≤75 μm; the ground sample was poured into a 32-mm mold and

**Table 1. Quantitative analysis results of elements in waste drilling fluid samples.**

| Elements | Ba | S | Ca | Si | Al | Fe | Others |
|---|---|---|---|---|---|---|---|
| Contents/% | 11.8 | 4.5 | 10 | 15 | 8 | 5 | 45.7 |

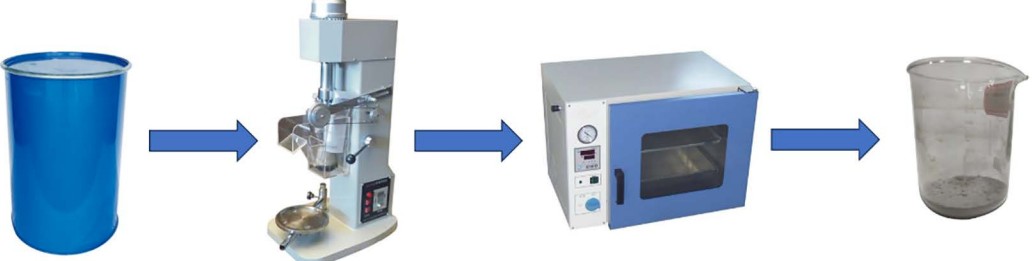

**Fig 1. Flow chart of the flotation experiment.**

pressed at 20 MPa for 30 seconds to form a circular sheet approximately 3 mm thick, which was then placed in the sample chamber. Quantitative analysis was conducted using the "standardless analysis mode" combined with a calibration curve; each sample was measured 3 times, and the average value was taken.

**2.3.3 Scanning electron microscopy (SEM) analysis.** The waste drilling fluid before flotation, as well as the barite and bentonite after flotation, were dried, ground, and passed through a 200 – mesh sieve. Subsequently, 3–5 g of each sample was taken, and the micro – morphological tests were carried out using a Hitachi S – 4800 field emission scanning electron microscope (FE – SEM).

**2.3.4 Contact angle measurement.** The pretreatment steps for the surface of waste drilling fluid samples are as follows: first, they are dispersed by ultrasonic vibration for 15 minutes, and then dried in a vacuum oven at 60°C for 2 hours. Afterwards, the samples are immersed in deionized water with a pH of 7.5 and a temperature of 25±1°C for 3 minutes, followed by sequential immersions in the required flotation reagent solutions (for concentrations, see Section 2.2). Each immersion in these solutions is carried out at a temperature of 25±1°C, with each immersion lasting 3 minutes. After each immersion, the surface of the sample needs to be carefully rinsed with deionized water (resistivity ≥18.2 MΩ‧ cm), and then dried again at 60°C for 1 hour before the test.

Contact angle measurements were performed using an optical wetting angle measuring instrument (OSA200) from Ningbo New Border Scientific Instruments Co., Ltd., China, under the following test conditions: The sessile drop method was adopted, with 5 µL of deionized water dispensed onto the sample surface via an automatic syringe; The temperature of the test chamber was maintained at 25±0.5°C, and the relative humidity was 50±3%; The contact angle values were recorded 30 seconds after the droplet stabilized to reduce the impact of evaporation;

Each sample was tested at 5 different positions, with 3 parallel measurements taken at each position. The final result was expressed as the average value±standard deviation (n = 15).

**2.3.5 Zeta potential measurement.** The dried waste drilling fluid samples were ground to a particle size of −2 µm and dispersed in a $10^{-3}$ M KCl solution to prepare a 0.01 wt% suspension. Flotation reagents were added according to the experimental requirements, and the pH was adjusted using HCl. After stirring and allowing the suspension to settle for 5 min, the supernatant was carefully collected for zeta potential analysis. Measurements were performed in triplicate under identical conditions, and the average value was reported as the final result.

## 3 Results and discussion

### 3.1 Optimization of oil-desorption reagents

To mitigate the adverse effects of oil-phase encapsulation on mineral surface hydrophobicity, a combination of ultrasonic treatment and various oil-desorption agents was employed for sample pretreatment, with dosages optimized accordingly. Experimental results (Fig 2) indicated that 0.5% sodium dodecyl sulfate (SDS) achieved the highest oil removal efficiency

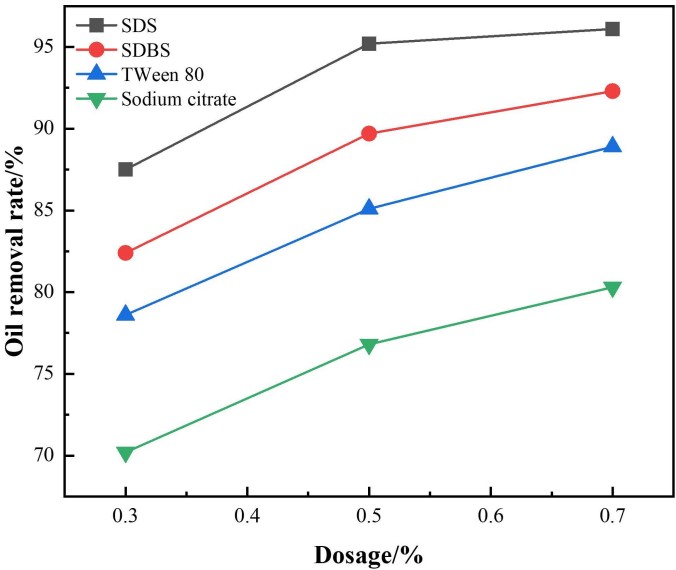

**Fig 2. Oil removal efficiency of different oil-desorption reagents.**

(95.2%) under ultrasonic assistance. The sulfonate groups in SDS effectively reduced interfacial tension and enhanced emulsification, outperforming sodium dodecylbenzene sulfonate (SDBS) and the bio-based surfactant Tween 80. This treatment successfully restored the natural hydrophilicity of mineral surfaces, creating favorable conditions for subsequent flotation. Although a higher SDS concentration (0.7%) slightly improved oil removal (96.1%), it resulted in a 10% decrease in energy efficiency and increased foam stability, rendering 0.5% the optimal dosage from an economic perspective.

### 3.2 Flotation reagent optimization

**3.2.1 First-stage flotation reagent optimization for bentonite.** With the collector fixed at 0.5 kg/t sodium dodecyl sulfate (SDS), the recovery efficiency of different depressants on bentonite in waste drilling fluid was investigated. Conversely, with the depressant fixed at 0.4 kg/t sodium hexametaphosphate, the recovery efficiency of different collectors on bentonite was evaluated. The results are shown in Figs 3 and 4, respectively.

As indicated in Fig 3, sodium hexametaphosphate exhibited significantly superior depression efficiency on bentonite, achieving a bentonite recovery rate of 52.7%. This is attributed to the $[PO_3]^{3-}$ groups chelating $Ba^{2+}$ on the barite surface to form a hydrophilic layer (evidenced by contact angle measurements). Fig 4 demonstrates that SDS at 0.5 kg/t achieved the optimal collection efficiency, with a bentonite recovery rate of 86.3%. The sulfonate groups ($-SO_3^-$) specifically adsorbed on edge $Al^{3+}$ to disrupt oil film encapsulation (contact angle increased by 105°). In contrast, dodecylamine (DDA) showed a recovery rate of only 78.5% due to pH limitation (required pH < 6.0), while sodium oleate suffered from precipitation caused by $Ca^{2+}$ interference (recovery rate plummeted to 52.1%), highlighting the universality of SDS in northern Shaanxi waste drilling fluids.

Based on the above analysis, using sodium hexametaphosphate as the depressant and SDS as the collector, laboratory optimization determined the optimal dosages for first-stage bentonite flotation as 0.8 kg/t and 0.6 kg/t, respectively.

**3.2.2 Second-stage flotation reagent optimization for barite.** Tailing samples from the first-stage flotation were collected for barite recovery via reverse flotation. With the pH fixed at 8.0 and 1.2 kg/t sodium dodecyl sulfate (SDS) added, the effect of gellan gum dosage on barite recovery from waste drilling fluid was investigated by adjusting gellan gum concentration, as shown in Fig 5.

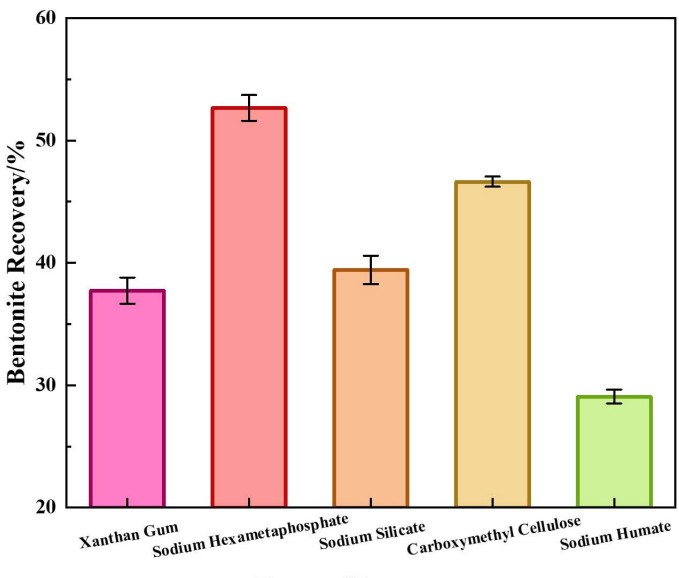

**Fig 3. Recovery efficiency of bentonite from waste drilling fluid with different depressants.**

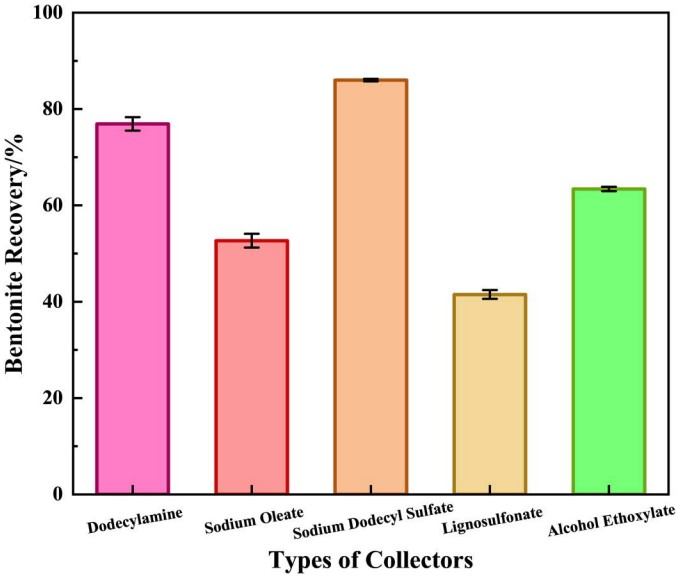

**Fig 4. Recovery efficiency of bentonite from waste drilling fluid with different collectors.**

Fig 5 reveals a significant antagonistic effect of gellan gum dosage on barite recovery and grade: when the dosage increased from 0.3 kg/t to 0.5 kg/t, barite recovery decreased from 92.5% to 88.7%, while the grade improved from 84.3% to 92.1%. This is primarily attributed to gellan gum selectively depressing bentonite flotation through carboxyl/hydroxyl hydrogen bonding [17]. Further increasing the dosage to 0.7 kg/t caused excessive gellan gum to coat the barite surface due to sharply increased viscosity, hindering sodium oleate adsorption and leading to a steep decline in recovery to

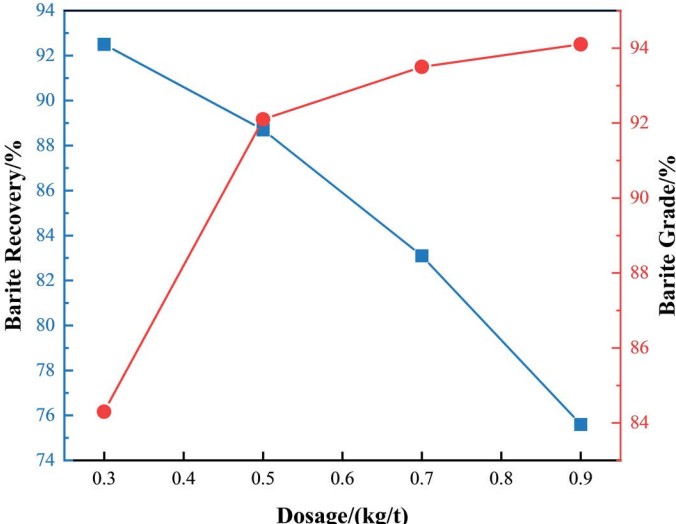

**Fig 5. Recovery efficiency of barite from waste drilling fluid with different dosages of gellan gum.**

83.1%. Although the grade slightly increased to 93.5%, the net benefit decreased by 12.9%. At 0.9 kg/t, the foam layer solidified, further deteriorating recovery to 75.6%.

In summary, 0.5 kg/t was determined as the optimal gellan gum dosage, balancing recovery (88.7%) and grade (92.1%) by achieving precise bentonite depression and moderate rheological control. This provides a critical regulatory threshold for the reverse flotation process of barite.

### 3.3 Closed-circuit flotation experiment

Based on the above flotation conditions, a "one roughing, two cleaning, three scavenging" flotation flow was adopted to conduct process exploration experiments, as shown in Figs 6 and 7. The results are presented in Tables 2 and 3.

In the first-stage bentonite flotation (using SDS as collector and sodium hexametaphosphate as depressant) via the closed-circuit process, bentonite concentrate with an overall recovery of 91.4% and grade of 91.5% was obtained. The three-stage cleaning process progressively increased montmorillonite purity from 78.3% (rough concentrate) to 86.2% (first cleaning) and 91.5% (second cleaning). Meanwhile, the three-stage scavenging process additionally recovered 7.3% low-grade bentonite ( > 54.7%), confirming the process efficiency in capturing fine particles.

For the second-stage barite reverse flotation (SDS as collector and gellan gum as depressant), the closed-circuit process achieved an overall barite recovery of 90.2% and grade of 92.1%. The cleaning stages removed residual bentonite, increasing $BaSO_4$ content by 7.8% (from 84.3% in rough concentrate to 92.1% in final concentrate). The scavenging system recovered 10.7% marginal-grade barite ( > 58.9%), significantly reducing resource loss.

SEM characterization (Fig 8) revealed the mineralogical features of concentrates: bentonite exhibited dissociated lamellar structures, while barite presented clean prismatic crystals without oil film encapsulation or impurity adhesion, verifying the separation effectiveness [18,19].

### 3.4 Flotation mechanism study

**3.4.1 Wettability.** The flotation mechanism was analyzed by investigating contact angle variations of bentonite and barite before/after collector/depressant addition, as shown in Table 4. Wettability studies revealed a regular antagonistic effect of reagents on surface hydrophobicity regulation: the original oil-contaminated bentonite exhibited a contact angle

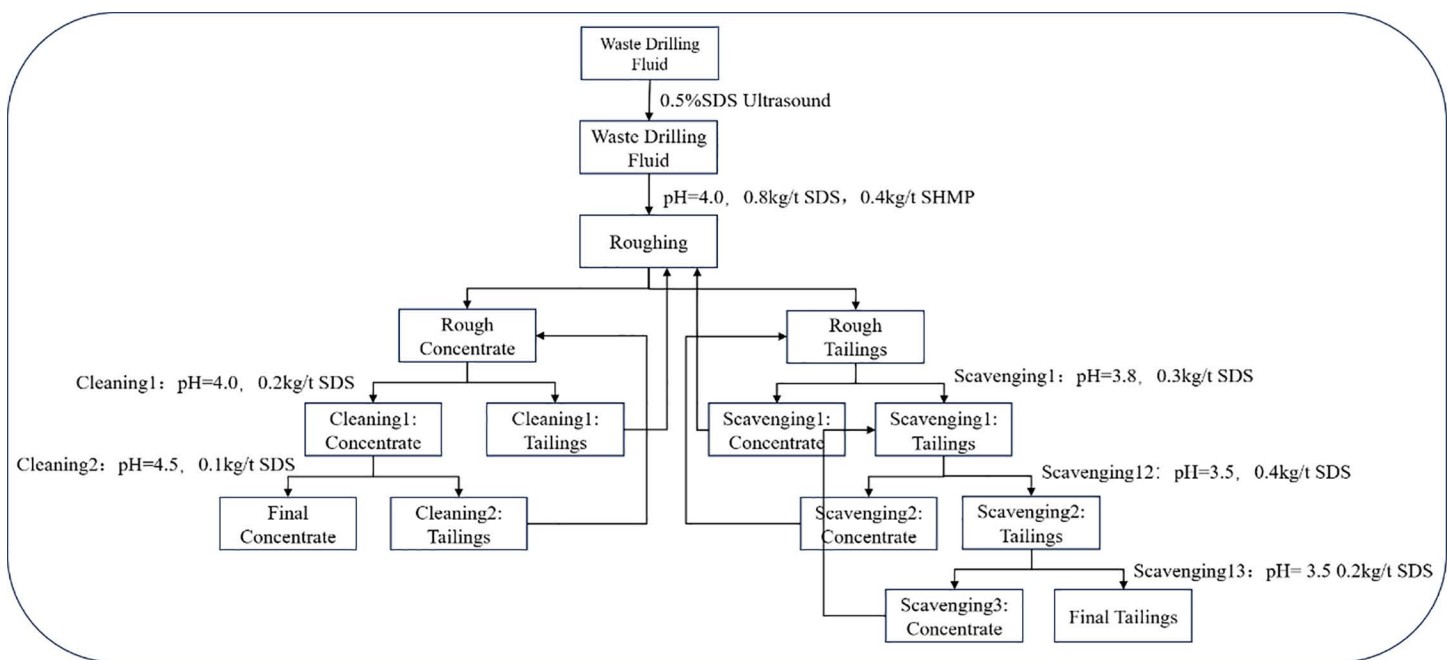

**Fig 6. Schematic diagram of bentonite flowsheet with "one roughing, two cleanings and three scavengings" in primary stage.**

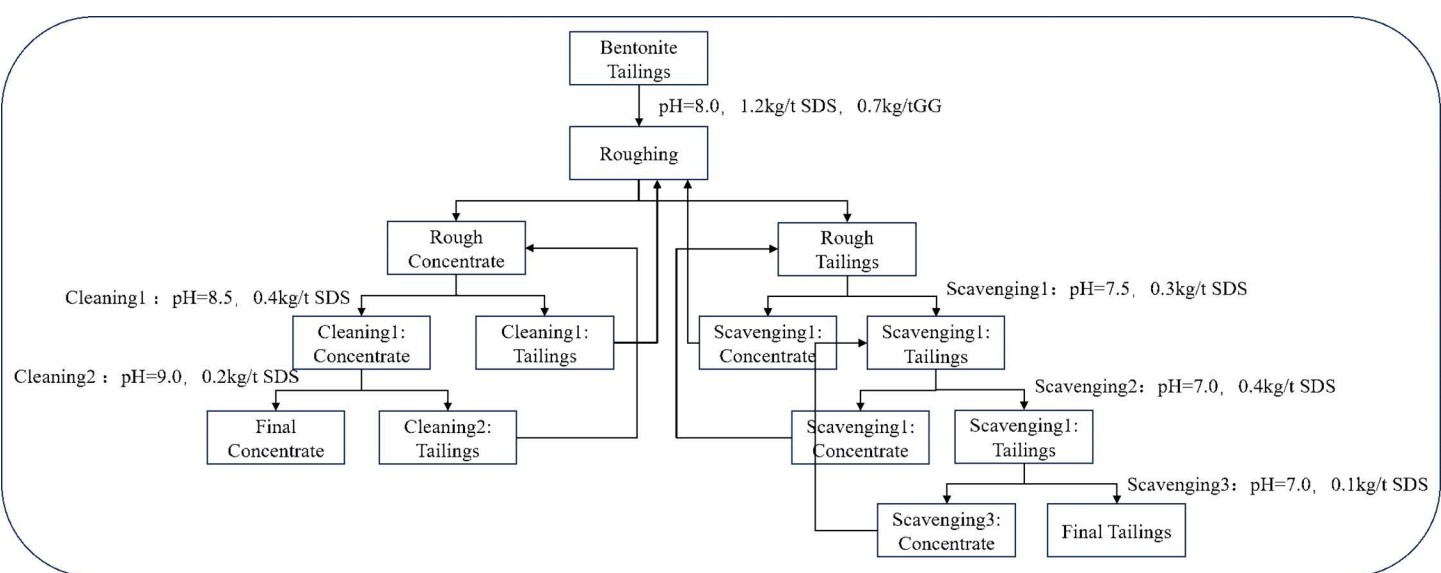

**Fig 7. Schematic diagram of barite flowsheet with "one roughing, two cleanings and three scavengings" in secondary stage.**

of 112° (strong hydrophobicity), which dropped sharply to 58° (hydrophilization) after cleaning with 0.5% SDS, confirming that sulfonate groups effectively stripped oil films to expose the silica-alumina hydrophilic surface. Upon adding 0.8 kg/t SDS as collector, the contact angle rebounded to 105° (strong hydrophobicity), attributed to the orientation of alkyl chains forming a hydrophobic layer after -$SO_3^-$ adsorbed on edge $Al^{3+}$. Conversely, 0.6 kg/t sodium hexametaphosphate deeply

**Table 2. Primary stage: bentonite "one roughing, two cleanings, three scavengings" results.**

| Stage | Product | Recovery/% | Grade/% |
|---|---|---|---|
| Roughing | Rough Concentrate | 92.5 | 78.3 |
| | Rough Tailings | 7.5 | / |
| Cleaning1 | Cleaning1:Concentrate | 85.7 | 86.2 |
| | Cleaning1: Tailings | 6.8 | 65.4 |
| Cleaning2 | Final Concentrate | 83.1 | 91.5 |
| | Cleaning2: Tailings | 2.6 | 72.1 |
| Scavenging1 | Scavenging1: Concentrate | 4.3 | 68.9 |
| Scavenging2 | Scavenging2: Concentrate | 2.1 | 61.2 |
| Scavenging3 | Scavenging3: Concentrate | 0.9 | 54.7 |
| Total Recovery | | 91.4 | / |

**Table 3. Secondary stage: barite "one roughing, two cleanings, three scavengings" results.**

| Stage | Product | Recovery/% | Grade/% |
|---|---|---|---|
| Roughing | Rough Concentrate | 88.7 | |
| | Rough Tailings | 11.3 | / |
| Cleaning1 | Cleaning1:Concentrate | 82.1 | 89.5 |
| | Cleaning1: Tailings | 6.6 | 70.8 |
| Cleaning2 | Final Concentrate | 80.5 | 92.1 |
| | Cleaning2: Tailings | 1.6 | 76.3 |
| Scavenging1 | Scavenging1: Concentrate | 5.2 | 73.6 |
| Scavenging2 | Scavenging2: Concentrate | 3.1 | 65.4 |
| Scavenging3 | Scavenging3: Concentrate | 1.4 | 58.9 |
| Total Recovery | | 90.2 | / |

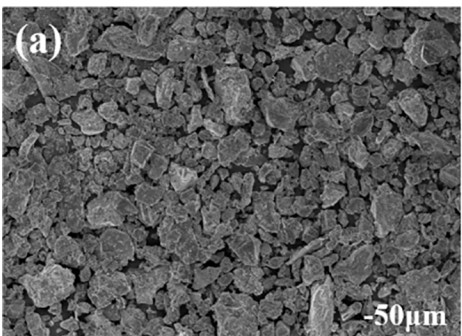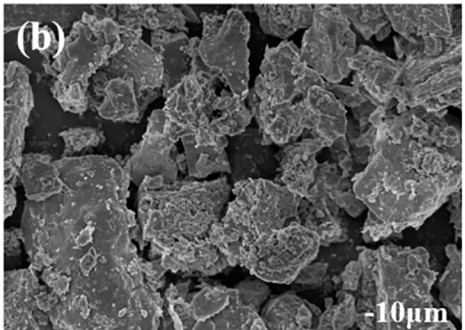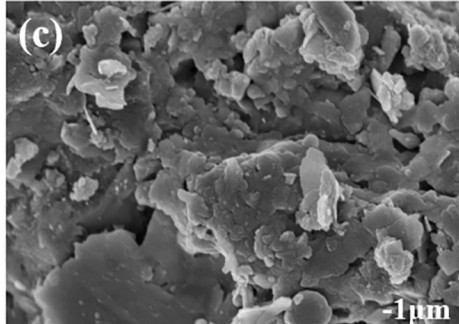

**Fig 8. SEM images of waste drilling fluid and recovered bentonite/barite ((a) 50μm; (b) 10μm; (c) 1μm).**

suppressed the contact angle to 38° (strong hydrophilicity) by chelating $Al^{3+}$ and introducing $[PO_3]^{3-}$ hydrophilic groups, revealing the micro-mechanism of depressants blocking collector adsorption by covering active sites [20–22].

For barite, the contact angle increased to 65° (weak hydrophobicity) at pH = 4.0 due to exposed $Ba^{2+}$, while 1.2 kg/t SDS achieved strong hydrophobicity (102°) via specific bonding of $-SO_3^-$ to $Ba^{2+}$. However, 0.5 kg/t gellan gum (GG) reduced

**Table 4. Contact angle changes of bentonite and barite before and after addition of collectors and depressants.**

| Mineral | Processing Conditions | Contact angle(°) | Hydrophobicity Variation |
|---|---|---|---|
| Bentonite | Pristine state | 112 | Strong hydrophobicity |
| | 0.5% SDS | 58 | Hydrophilization |
| | 0.5% SDS + 0.8 kg/t SDS | 105 | Strong hydrophobicity |
| | 0.5% SDS + 0.6 kg/t SHMP | 38 | Strong Hydrophilicity |
| Barite | Pristine state | 45 | Weak Hydrophilicity |
| | pH = 4.0 | 65 | Weak Hydrophobicity |
| | 1.2 kg/t SDS | 102 | Strong hydrophobicity |
| | 0.5 kg/t GG | 41 | Strong Hydrophilicity |

the contact angle to 41° (strong hydrophilicity) by constructing a hydration film through -COOH/-OH groups, verifying its steric hindrance depressing efficiency.

**3.4.2 Zeta potential.** The results of zeta potential changes of bentonite and barite before and after the addition of collector and depressant are shown in Table 5. The Zeta potential evolution in the bentonite flotation system was dominated by the synergistic effect of anionic reagents: In the first stage, the sulfonate groups ($-SO_3^-$) of sodium dodecyl sulfate (SDS) preferentially adsorbed onto edge $Al^{3+}$ sites of bentonite under acidic conditions, shifting the potential from −18 mV (oil-contaminated state) to −42 mV. Concurrently, the alkyl chains ($C_{12}H_{25}-$) oriented to form a hydrophobic layer, driving the contact angle to 98° for bentonite collection. The depressant sodium hexametaphosphate strongly chelated interlayer $Ca^{2+}/Al^{3+}$ via phosphate groups ($[PO_3]^{3-}$), not only competitively dissociating adsorbed SDS but also further pushing the potential to −52 mV, constructing an electrical double layer hydration barrier on the mineral surface. This caused the contact angle to plummet to 42°, completely suppressing residual bentonite flotation [23].

During the subsequent barite flotation, SDS acted as a collector again, adsorbing specifically to $Ba^{2+}$ via - $SO_3^-$, reducing the potential from −22 mV to −38 mV and hydrophobizing barite (contact angle 102°). However, the depressant gellan gum ionized its carboxyl/hydroxyl groups (-COOH/-OH) at high pH, generating super negative potential (−45 mV). Its polymer chains coated the barite surface through hydrogen bonding and steric hindrance, both shielding SDS adsorption sites and forming a hydrophilic film, depressing the contact angle below 45° to ensure barite concentrate purity [24].

The entire process reveals a dual potential control mechanism: bentonite flotation relies on the dynamic antagonism between "SDS negative-charge hydrophobization" and "sodium hexametaphosphate super-negative inhibition", while barite recovery depends on the competitive balance of "secondary SDS hydrophobic adsorption" and "gellan gum steric shielding".

**Table 5. Zeta potential changes of bentonite and barite before and after addition of collector and depressant.**

| Mineral | Processing Conditions | Zeta Potential/mV |
|---|---|---|
| Bentonite | Pristine state | −18 |
| | 0.5% SDS | −42 |
| | 0.5% SDS + 0.8 kg/t SDS | 15 |
| | 0.5% SDS + 0.6 kg/tSHMP | −52 |
| Barite | Pristine state | −22 |
| | pH = 4.0 | −5 |
| | 1.2 kg/tSDS | −38 |
| | 0.5 kg/t GG | −45 |

### 3.4.3 Comprehensive mechanism analysis.

The schematic diagram of valuable solid phase recovery mechanism from waste drilling fluid is shown in Fig 9. The essence of stepwise flotation for bentonite-barite lies in the "charge-wettability" directional reconstruction induced by SDS at the oil-mineral interface and the competitive decoupling process of depressants: In the first-stage bentonite flotation, the sulfonate groups ($-SO_3^-$) of SDS form coordinate bonds with edge $Al^{3+}$ of bentonite, triggering charge reversal ($\zeta$: $-42$ mV → $+15$ mV) and self-assembly of alkyl chains into a hydrophobic layer (contact angle 105°), enabling selective collection of oil-encapsulated minerals and exposing the mineral surface. Sodium hexametaphosphate competitively desorbs SDS via high-affinity phosphate groups, simultaneously shifting the potential to a deep negative value of $-52$ mV to form an electrical double layer hydration barrier (contact angle 38°), completely suppressing residual bentonite [25–28].

In the subsequent barite flotation stage, SDS specifically adsorbs onto $Ba^{2+}$ on the barite surface. Although the potential shifts negatively ($\zeta$: $-22$ mV → $-38$ mV), the hydrophobic stacking of alkyl chains (contact angle 102°) overcomes electrostatic constraints for flotation. Gellan gum constructs high-molecular-weight steric micelles (particle size ≈85 nm) through carboxyl ionization and hydrogen bond networks, blocking the adsorption pathway of SDS and achieving characteristic separation of bentonite and barite during flotation.

## 4 Conclusions

(1) Ultrasonic cleaning with a 0.5% sodium dodecyl sulfate (SDS) solution achieved a solid-phase oil desorption rate >95%. Using a stepwise flotation process, the first stage at pH=4.0 with 0.8 kg/t SDS and 0.6 kg/t sodium hexametaphosphate yielded a bentonite recovery rate of 86.3%. The second stage at pH=8.0 with 1.2 kg/t SDS and 0.7 kg/t gellan gum produced barite concentrate with a grade of 92.1% ($BaSO_4$ content) and a recovery rate of 88.7%.

(2) Based on the "one roughing, two cleaning, three scavenging" 流程，the first-stage bentonite flotation (SDS+sodium hexametaphosphate) achieved a concentrate with 91.4% recovery and 91.5% grade. The three-stage cleaning

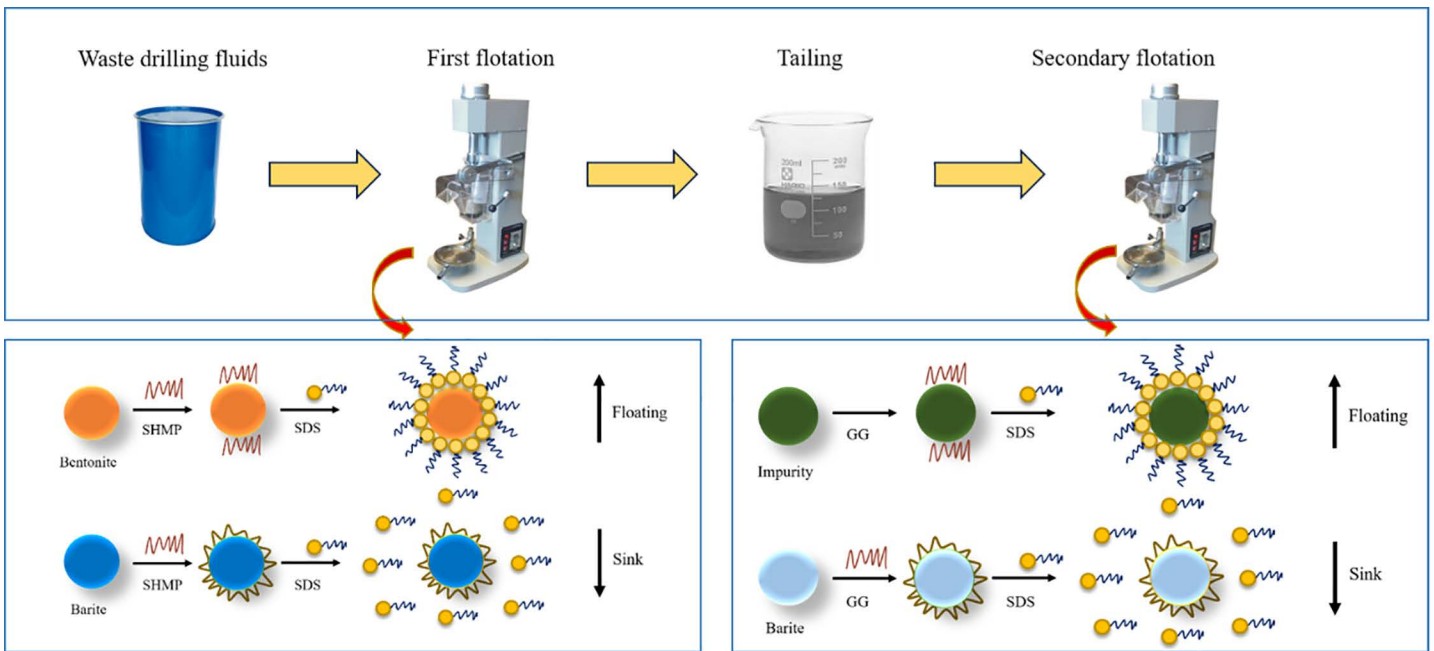

**Fig 9. Schematic diagram of valuable solid phase recovery mechanism from waste drilling fluid.**

process increased purity by 13.2%, and scavenging recovered 7.3% low-grade material. The second-stage barite flotation (SDS + gellan gum) reached 90.2% recovery and 92.1% grade. SEM images revealed dissociated lamellar bentonite and clean prismatic barite crystals without oil films or impurities, verifying the high efficiency of the separation.

(3) Mechanistic studies indicated that SDS exhibits dual functions of oil film disruption and collection. Sodium hexametaphosphate inhibits barite flotation by chelating $Ba^{2+}$, while gellan gum achieves separation by shielding active sites on bentonite via hydrogen bonding. This research provides valuable insights for the resource utilization of drilling waste fluids in northern Shaanxi.

(4) This study has not systematically evaluated the large-scale economic viability of the stepwise flotation process; the laboratory data on reagent consumption and energy consumption are difficult to match industrial operating conditions, and the adaptability and cost control of the process for waste drilling fluids with high oil content or complex compositions remain to be verified.

## Author contributions

**Conceptualization:** Shi Jiahe.

**Data curation:** Shi Jiahe.

**Formal analysis:** Shi Jiahe.

**Investigation:** Liu Lihu, Shi Jiahe.

**Project administration:** Shi Jiahe.

**Resources:** Shi Jiahe.

**Supervision:** Shi Jiahe.

**Writing – original draft:** Liu Lihu.

**Writing – review & editing:** Liu Lihu.

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
