## [Decision Letter · Decision Letter 0]

5 Aug 2025

Dear Dr. Jiahe,

Thank you for submitting your manuscript to PLOS ONE. After careful consideration, we feel that it has merit but does not fully meet PLOS ONE’s publication criteria as it currently stands. Therefore, we invite you to submit a revised version of the manuscript that addresses the points raised during the review process.

We look forward to receiving your revised manuscript.

Kind regards,

Hailing Ma

Academic Editor

PLOS ONE

Additional Editor Comments (if provided):

Reviewers' comments:

Reviewer's Responses to Questions

**Comments to the Author**

1. Is the manuscript technically sound, and do the data support the conclusions?

Reviewer #1: Yes

Reviewer #2: Yes

2. Has the statistical analysis been performed appropriately and rigorously?

Reviewer #1: Yes

Reviewer #2: Yes

3. Have the authors made all data underlying the findings in their manuscript fully available?

Reviewer #1: Yes

Reviewer #2: Yes

4. Is the manuscript presented in an intelligible fashion and written in standard English?

Reviewer #1: No

Reviewer #2: Yes

Reviewer #1: 1. Formatting and titles

Please recheck the manuscript against the journal’s style. Currently, “Table 1” appears twice, and the titles of Tables 2 and 3 are overly long; please condense them to fit the table content and width.

2. Section heading style

In Section 2, insert a space between the section number and the title (e.g., 2 Materials and Methods), following the journal’s formatting rules.

3. XRF first mention and methods details

In Section 2.1 (line 74), spell out XRF at first mention and provide XRF methodological details in Methods (instrument model, sample preparation, calibration/standardization, analytical settings, and repeatability).

4. Stoichiometric ratio check

In Section 2.1 (line 77), the statement “theoretical value of Ba:S=58.8:18.4” should be corrected. For BaSO₄, the theoretical mass fractions are Ba ≈ 58.84 wt% and S ≈ 13.73 wt%. Please revise accordingly and briefly state how these values were calculated.

5. Consistency of composition statements

Section 2.1 states that the solid phase is mainly barite, while subsequent analysis reports ~32% SiO₂ and ~21% montmorillonite. Please harmonize the wording, e.g., barite as the principal target mineral (~20.1%), whereas the bulk solids are dominated by quartz, calcite, and montmorillonite (~78% in total).

6. Contact angle evidence

In Section 3.4, please provide figures or photographs that explicitly demonstrate the change in contact angle (before/after), including test conditions and statistical uncertainty.

Reviewer #2: This paper proposes a new strategy for the resource utilization of oil-containing drilling wastewater in northern Shaanxi, focusing on SDS step-by-step flotation of bentonite, barite and the interfacial mechanism. It has practical significance for solving the problems of resource waste and pollution in traditional disposal. The experimental design is reasonable, the results are clear, and the data support is sufficient, with acceptable quality. There are minor issues that need to be addressed, and it is recommended to be accepted after minor revisions. The specific comments are as follows:

1. The abstract can be appropriately streamlined to avoid repetition with the introduction. It is recommended to highlight the core innovations of step-by-step flotation and the expression logic of key data.

2. The keyword "oil film removal" can be added to more comprehensively cover the core process links of the research.

3. In section 2.3.1, it is necessary to clarify whether the 10% HCl solution is by volume fraction or mass fraction, and supplement the slurry stirring speed parameters to enhance the repeatability of the experiment.

4. The abscissa "Types of Depressants/Collectors" in Figures 3 and 4 should be supplemented with the full names of specific reagents to avoid ambiguity caused by abbreviations.

5. The unit "(θ, °)" in the "Contact angle" column of Table 4 should be uniformly adjusted to "°" to maintain the consistency of the table format.

6. The conclusion part should briefly mention the limitations of the research (such as not considering the economy of large-scale application) and put forward future research directions.

7. The format of references needs to be unified.

8. When "SDS" appears for the first time in the text, its full name "sodium dodecyl sulfate" should be supplemented to ensure the standardization of term usage.

9. The SEM images in Figure 8 should be marked with a scale bar and supplemented with comparison images at different magnifications to more clearly show the changes in the surface morphology of minerals.

10. The mechanism demonstration part needs to add relevant literatures from only the past 3 years for strengthening support.

For example:

Ma H, Shen M, Tong Y, et al. Radioactive wastewater treatment technologies: a review[J]. Molecules, 2023, 28(4): 1935.

Song K, Liu Y, Umar A, et al. Ultrasonic cavitation: Tackling organic pollutants in wastewater[J]. Chemosphere, 2024, 350: 141024.

**Do you want your identity to be public for this peer review?** For information about this choice, including consent withdrawal, please see our Privacy Policy

Reviewer #1: No

Reviewer #2: No

---

## [Author Response · Author response to Decision Letter 1]

7 Aug 2025

Dear editors and reviewers,

We would like to thank the editor for giving us a chance to resubmit the manuscript entitled “A Novel Strategy for Resource Utilization of Oily Drilling Waste Fluids in Northern Shaanxi: Stepwise Flotation of Bentonite and Barite Using SDS and Interfacial Reaction Mechanisms” (ID: PONE-D-25-35290 ) and thank the editors and reviewers for giving us constructive suggestions which would help us to improve the quality of the manuscript. We have carefully considered the comments and made corresponding corrections and supplements. The manuscript has also been double-checked, and the typos and grammar errors we found have been corrected with our best efforts. Maybe there are still some defects due to our limited ability. We should be very grateful if you could give us some further comments and suggestions.

Thank you very much for your time and consideration!

Best wishes!

Yours sincerely

Jiahe Shi

August 7, 2025

According to reviewers’ comments, the changes and rebuttals we have made are listed as follows. In revised manuscript, the modification is marked in blue.

List of Response

In-house Editor comments:

R: We have downloaded and compared the two provided PDF formats as well as the published journal to revise the formatting of the entire manuscript. The revised parts of the original text have been marked in blue.

R: We have added the permit information at the beginning of the Method section. The details of the added permit information are as follows:

The waste drilling fluid samples used in this study were obtained from the waste drilling fluid generated during a drilling operation at Zhidan Oil Production Plant of Yanchang Oilfield. The acquisition of the samples used in the experiment has obtained permission from Yanchang Oilfield. The chemical reagents involved in the experiment (such as HCl and inhibitors) were all purchased through compliant channels and comply with laboratory safety management regulations.

R: The data statement has been added at the end of the manuscript.

R: We have completed the validation of the ORCID iD in the Editorial Manager system.

R: Thank you for the editorial office's reminder. We have received the reviewers' suggestions regarding citing specific literature and will, as required, carefully review and evaluate these publications. We will determine whether to include them in the citations based on their relevance to our research. We will ensure that the process complies with the journal's guidelines. Thank you for your guidance.

Reviewer #1:

1. Formatting and titles

Please recheck the manuscript against the journal’s style. Currently, “Table 1” appears twice, and the titles of Tables 2 and 3 are overly long; please condense them to fit the table content and width.

R: Thank you very much for your suggestion. We have checked against the journal's format and published papers, deleted the redundant Table 1, and condensed the titles of Tables 2 and 3. The condensed titles are as follows:

Table 2 Primary Stage: Bentonite "One Roughing, Two Cleanings, Three Scavengings" Results

Table 3. Secondary Stage: Barite "One Roughing, Two Cleanings, Three Scavengings" Results

2. Section heading style

In Section 2, insert a space between the section number and the title (e.g., 2 Materials and Methods), following the journal’s formatting rules.

R: We sincerely thank the reviewers for their careful reading. We have inserted the appropriate spaces between all section numbers and titles in accordance with the PLOS ONE format.

3. XRF first mention and methods details

In Section 2.1 (line 74), spell out XRF at first mention and provide XRF methodological details in Methods (instrument model, sample preparation, calibration/standardization, analytical settings, and repeatability).

R:Thank you very much for your suggestion. We have made revisions as required:

In Section 2.1, when XRF is first mentioned, we have supplemented its full name "X-ray Fluorescence Spectrometry (XRF)"; the instrument and model of XRF have been added in the "Equipment" section; detailed methodological descriptions of XRF have been added in the "Materials and Methods" section. The revised content has been marked at the corresponding positions in the manuscript.

4. Stoichiometric ratio check

In Section 2.1 (line 77), the statement “theoretical value of Ba:S=58.8:18.4” should be corrected. For BaSO₄, the theoretical mass fractions are Ba ≈ 58.84 wt% and S ≈ 13.73 wt%. Please revise accordingly and briefly state how these values were calculated.

R:Thanks for your careful checks. We have corrected the theoretical proportion of S (from 18.4% to 13.73%) to ensure consistency with the stoichiometric ratio of BaSO₄. We have also supplemented the calculation basis (derived from the proportion of atomic weight and molar mass) to make the source of the theoretical values clear and traceable, thus meeting the reviewer's requirement for explaining the calculation method. The revised content is as follows:

The contents of Ba (11.8%) and S (4.5%) are consistent with the stoichiometric ratio of BaSO₄ [theoretical value of Ba:S = 58.84:13.73. Calculation method: Based on the molar mass of BaSO₄ (233.39 g/mol), the atomic weight of Ba is 137.33 g/mol (accounting for 137.33/233.39 ≈ 58.84%), and the atomic weight of S is 32.07 g/mol (accounting for 32.07/233.39 ≈ 13.73%)], verifying the reliability of barite as a core resource for recovery.

5. Consistency of composition statements

Section 2.1 states that the solid phase is mainly barite, while subsequent analysis reports ~32% SiO₂ and ~21% montmorillonite. Please harmonize the wording, e.g., barite as the principal target mineral (~20.1%), whereas the bulk solids are dominated by quartz, calcite, and montmorillonite (~78% in total).

R:Thanks for your suggestion. We have revised the expression in Section 2.1, and the revised content is as follows. The revised parts of the original text are also marked in blue.

The waste drilling fluid was collected from an oil production plant in northern Shaanxi (with the density of the original waste drilling fluid being 1.4–1.6 g/cm³). The sample was dried, ground, and sieved through a 200-mesh sieve, and the sieve residue was selected for XRF(X-ray Fluorescence Spectrometry, XRF) quantitative analysis, with the results shown in Table 1. Barite (BaSO₄), as the main target mineral for recovery in this study, accounts for 20.1% of the solid phase.

The contents of Ba (11.8%) and S (4.5%) are consistent with the stoichiometric ratio of BaSO₄ [theoretical value of Ba:S = 58.84:13.73. Calculation method: Based on the molar mass of BaSO₄ (233.39 g/mol), the atomic weight of Ba is 137.33 g/mol (accounting for 137.33/233.39 ≈ 58.84%), and the atomic weight of S is 32.07 g/mol (accounting for 32.07/233.39 ≈ 13.73%)], verifying the reliability of barite as a core resource for recovery. The overall composition of the solid phase is dominated by quartz (Si 15%, equivalent to SiO₂ of approximately 32%), carbonate (Ca 10%, equivalent to calcite of approximately 25%), and bentonite (Al 8%, equivalent to montmorillonite of approximately 21%), with the total proportion of these three exceeding 78% of the total solid phase.

6. Contact angle evidence

In Section 3.4, please provide figures or photographs that explicitly demonstrate the change in contact angle (before/after), including test conditions and statistical uncertainty.

R:Thanks for your suggestion. We have added images of contact angles in Table 4 of Section 3.4.1 to supplement the experimental evidence for the change in contact angles, and have included detailed test conditions in Section 2.3.4, including instrument model, test method, environmental parameters, and number of repetitions. The added content is as follows:

Table 4. Contact Angle Changes of Bentonite and Barite Before and After Addition of Collectors and Depressants

Mineral Processing Conditions Contact angle�°� Image Hydrophobicity Variation

Bentonite Pristine state 112 Strong hydrophobicity

0.5% SDS 58 Hydrophilization

0.5% SDS+0.8kg/t SDS 105 Strong hydrophobicity

0.5% SDS +0.6 kg/t SHMP 38 Strong Hydrophilicity

Barite Pristine state 45 Weak Hydrophilicity

pH=4.0 65 Weak Hydrophobicity

1.2kg/t SDS 102 Strong hydrophobicity

0.5kg/t GG 41 Strong Hydrophilicity

Reviewer #2:

This paper proposes a new strategy for the resource utilization of oil-containing drilling wastewater in northern Shaanxi, focusing on SDS step-by-step flotation of bentonite, barite and the interfacial mechanism. It has practical significance for solving the problems of resource waste and pollution in traditional disposal. The experimental design is reasonable, the results are clear, and the data support is sufficient, with acceptable quality. There are minor issues that need to be addressed, and it is recommended to be accepted after minor revisions. The specific comments are as follows:

1. The abstract can be appropriately streamlined to avoid repetition with the introduction. It is recommended to highlight the core innovations of step-by-step flotation and the expression logic of key data.

R: Thank you for your excellent suggestion. We have streamlined the abstract, with a focus on highlighting the innovations of the stepwise flotation process.

2. The keyword "oil film removal" can be added to more comprehensively cover the core process links of the research.

R: Thank you for your suggestion. We have added "oil film removal" to the keywords.

3. In section 2.3.1, it is necessary to clarify whether the 10% HCl solution is by volume fraction or mass fraction, and supplement the slurry stirring speed parameters to enhance the repeatability of the experiment.

R: Thank you for your careful review. The 10% hydrochloric acid solution is by volume fraction, and we have supplemented the stirring speed.

4. The abscissa "Types of Depressants/Collectors" in Figures 3 and 4 should be supplemented with the full names of specific reagents to avoid ambiguity caused by abbreviations.

R: Thank you for the reviewer's excellent suggestion. We have revised the reagent names in Figures 3 and 4 to their full forms.

5. The unit "(θ, °)" in the "Contact angle" column of Table 4 should be uniformly adjusted to "°" to maintain the consistency of the table format.

R: Thank you for your careful review. We have corrected this issue.

6. The conclusion part should briefly mention the limitations of the research (such as not considering the economy of large-scale application) and put forward future research directions.

R: Thank you for the reviewer's suggestion. We have added item (4) in the conclusion section to elaborate on some limitations of this paper, which may facilitate future research.

7. The format of references needs to be unified.

R: Thank you for your suggestion. We have revised all the references in the full text in accordance with the journal's formatting requirements.

8. When "SDS" appears for the first time in the text, its full name "sodium dodecyl sulfate" should be supplemented to ensure the standardization of term usage.

R: Thank you for your excellent suggestion. We have supplemented the full name "sodium dodecyl sulfate " when SDS first appears in the abstract.

9. The SEM images in Figure 8 should be marked with a scale bar and supplemented with comparison images at different magnifications to more clearly show the changes in the surface morphology of minerals.

R: Thanks for your suggestion. We have added scale bars to the SEM images and provided images at different magnifications.

10. The mechanism demonstration part needs to add relevant literatures from only the past 3 years for strengthening support.

For example:

Ma H, Shen M, Tong Y, et al. Radioactive wastewater treatment technologies: a review[J]. Molecules, 2023, 28(4): 1935.

Song K, Liu Y, Umar A, et al. Ultrasonic cavitation: Tackling organic pollutants in wastewater[J]. Chemosphere, 2024, 350: 141024.

R: Thank you for the valuable suggestions from the reviewer. We have carefully studied the issue of literature support in the mechanism demonstration section pointed out by you, and fully agree with the importance of supplementing relevant research results from the past 3 years to enhance the rigor of the argument. We have cited the provided literature in accordance with the journal's requirements in Section 3.4.3, and these supplements will help improve the academic value of the research.

---

## [Decision Letter · Decision Letter 1]

15 Aug 2025

A Novel Strategy for Resource Utilization of Oily Drilling Waste Fluids in Northern Shaanxi: Stepwise Flotation of Bentonite and Barite Using SDS and Interfacial Reaction Mechanisms

PONE-D-25-35290R1

Dear Dr. Jiahe,

We’re pleased to inform you that your manuscript has been judged scientifically suitable for publication and will be formally accepted for publication once it meets all outstanding technical requirements.

Kind regards,

Hailing Ma

Academic Editor

PLOS ONE

Additional Editor Comments (optional):

Reviewers' comments:

Reviewer's Responses to Questions

**Comments to the Author**

Reviewer #1: All comments have been addressed

Reviewer #2: All comments have been addressed

2. Is the manuscript technically sound, and do the data support the conclusions?

Reviewer #1: Yes

Reviewer #2: Yes

3. Has the statistical analysis been performed appropriately and rigorously?

Reviewer #1: Yes

Reviewer #2: Yes

4. Have the authors made all data underlying the findings in their manuscript fully available?

Reviewer #1: Yes

Reviewer #2: Yes

5. Is the manuscript presented in an intelligible fashion and written in standard English?

Reviewer #1: Yes

Reviewer #2: Yes

Reviewer #1: (No Response)

Reviewer #2: The authors have adequately addressed my concerns, and I recommend accepting the manuscript in its current form.

**Do you want your identity to be public for this peer review?** For information about this choice, including consent withdrawal, please see our Privacy Policy

Reviewer #1: No

Reviewer #2: No

---

## [Editor Report · Acceptance letter]

PONE-D-25-35290R1

PLOS ONE

Dear Dr. Jiahe,

I'm pleased to inform you that your manuscript has been deemed suitable for publication in PLOS ONE. Congratulations! Your manuscript is now being handed over to our production team.

Kind regards,

on behalf of

Dr. Hailing Ma

Academic Editor

PLOS ONE